# A systematic review of trial registry entries for randomized clinical trials investigating COVID-19 medical prevention and treatment

Anders Peder Højer Karlsen[1]*, Sebastian Wiberg[1], Jens Laigaard[1], Casper Pedersen[1], Kim Zillo Rokamp[1], Ole Mathiesen[1,2]

1 Center for Anaesthesiological Research, Department of Anaesthesiology, Zealand University Hospital, Koege, Denmark, 2 Department of Clinical Medicine, University of Copenhagen, Copenhagen, Denmark

* andersphkarlsen@gmail.com

## Abstract

### Aim

To identify investigated interventions for COVID-19 prevention or treatment via trial registry entries on planned or ongoing randomised clinical trials. To assess these registry entries for recruitment status, planned trial size, blinding and reporting of mortality.

### Methods

We identified trial registry entries systematically via the WHO International Clinical Trials Registry Platform and 33 trial registries up to June 23, 2020. We included relevant trial registry entries for randomized clinical trials investigating medical preventive, adjunct or supportive therapies and therapeutics for treatment of COVID-19. Studies with non-random and single-arm design were excluded. Trial registry entries were screened by two authors independently and data were systematically extracted.

### Results

We included 1303 trial registry entries from 71 countries investigating 381 different single interventions. Blinding was planned in 47% of trials. Sample size was >200 participants in 40% of trials and a total of 611,364 participants were planned for inclusion. Mortality was listed as an outcome in 57% of trials. Recruitment was ongoing in 54% of trials and completed in 8%. Thirty-five percent were multicenter trials. The five most frequent investigational categories were immune modulating drugs (266 trials (20%)), unconventional medicine (167 trials (13%)), antimalarial drugs (118 trials (9%)), antiviral drugs (100 trials (8%)) and respiratory adjuncts (78 trials (6%)). The five most frequently tested uni-modal interventions were: chloroquine/hydroxychloroquine (113 trials with 199,841 participants); convalescent plasma (64 trials with 11,840 participants); stem cells (51 trials with 3,370 participants); tocilizumab (19 trials with 4,139 participants) and favipiravir (19 trials with 3,210 participants).

**Data Availability Statement:** All relevant data are within the paper and its Supporting Information file.

**Funding:** The authors received no specific funding for this work.

**Competing interests:** The authors have declared that no competing interests exist.

## Conclusion

An extraordinary number of randomized clinical trials investigating COVID-19 management have been initiated with a multitude of medical preventive, adjunctive and treatment modalities. Blinding will be used in only 47% of trials, which may have influence on future reported treatment effects. Fifty-seven percent of all trials will assess mortality as an outcome facilitating future meta-analyses.

## Background

The novel corona virus (SARS-CoV-2) outbreak began in late December 2019 and rapidly spread across the globe critically impacting public health systems. SARS-CoV-2 is causing respiratory disease (COVID-19) ranging from asymptomatic cases and mild symptoms of upper airway infection to fulminant acute respiratory distress syndrome (ARDS), multi-organ failure and death [1]. This accelerating pandemic has inclined researchers around the world to accelerate investigative efforts to find effective and safe COVID-19 prevention and treatment options.

On January 30, 2020 the Emergency Committee convened by the World Health Organization (WHO) declared the COVID-19 outbreak a Public Health Emergency of International Concern (PHEIC) and has presented a collaborative research agenda with eight defined areas that should be prioritized during the early phase of the pandemic [2, 3]. The first wave of published evidence for treatment modalities relied primarily on observational studies and indirect evidence from adjacent fields [4]. Now countries and authorities have established modified fast track pathways to ethical and other approvals [5, 6], and journals have fast track publication processes, all enabling rapid conduction and publication of randomized clinical trials (RCTs) to ensure a second wave of high-quality evidence with proper assessment of benefits and harms [7, 8] (preprint). WHO has created a Global Research Roadmap homepage to facilitate and oversee that critical research is prioritized and implemented in the correct order [9].

Even though multiple versions of trial registration databases are readily available for download online [10–12], trialists may find it difficult to get an overview of the constantly evolving multitude of planned or ongoing studies. Therefore, we aim to provide a global snapshot overview of interventions and main methodological aspects of planned and initiated randomised clinical trials on COVID-19 prevention and treatment. To do so, we assess available trial registry entries from 33 clinical trial registries to June 23, 2020.

## Methods

For this topical review, we identified trial registry entries from the datafile of June 23, 2020 provided by the COVID-19 section of the WHO International Clinical Trials Registry Platform (ICTRP), which included information from 10 different trial registries [13]. Further, we systematically searched these and 23 other national and international trial registries (S1 Appendix) that are listed in the Cochrane recommended list of Clinical Trial Registers provided by York Health Economics Consortium, University of York [14]. We searched on Title: "corona OR COVID OR sars-cov-2". The final search was performed on June 23, 2020.

We included trial registry entries on randomized clinical trials investigating medical preventive, adjunct or supportive therapies and therapeutics for treatment of COVID-19. We included trial registry entries irrespective of participant age. Trials with non-random and single-arm designs were excluded. Trials assessing non-medical interventions were excluded.

Trial registry entries were screened for inclusion and data were extracted by two authors independently (KZR or APHK and JL or OM) and discrepancies were handled by a senior author (APHK or OM). Data were extracted into Excel and tables were generated using formatted coding.

## Outcomes

The primary outcome was to identify and explore COVID-19 prevention or treatment interventions in planned or ongoing randomized clinical trials. The secondary outcomes were: 1) recruitment status; 2) planned trial size; 3) blinding; 4) reporting of mortality as a primary or secondary outcome.

## Data handling

Data on type of intervention, mortality assessment, blinding, trial size, multicenter registration, recruitment status and disease severity were extracted as it was reported in the trial registry entries. Mortality was extracted binary regardless of cause-specificity or timely differences. The clinical research phase was registered (phase 1–4 trials) and whenever an entry had multiple trial phases, we registered the highest. We reported the number of blinded parties registered in the trial registry entries, but not who was blinded. Disease severity was subsequently categorized as either prevention, mild/moderate and severe/hospitalized/ICU. Recruitment status was updated up to June 23, 2020. Trial registry entries were categorized into intervention subgroups based on therapeutic class or mechanism of action. Data are presented as numbers and percentages.

## Results

We identified 4,040 trial registry entries and included 1,303 registered RCTs, originating from 71 different countries on 6 continents (Fig 1 and Table 1 and S2 and S3 Appendices). The vast majority of exclusions were due to single-group, observational and non-randomised designs, duplicates and testing of other interventions than COVID-19 treatment or prevention. Our database is available as a .csv file (S4 Appendix).

### Interventional characteristics

The 1303 trial registry entries included 381 different single interventions and 126 combinations or multiple comparisons (S3 Appendix).

Trial registry entries were categorized into 18 different therapeutic groups (Tables 1–3). The five most frequent investigational categories were immune modulating drugs (266 trials (20%)), unconventional medicine (167 trials (13%)), antimalarial drugs (118 trials (9%)), antiviral drugs (100 trials (8%)) and respiratory adjuncts (78 trials (6%)).

Twenty-two specific therapeutic medical interventions with seven or more trial registry entries per intervention were described separately (Table 4). The five most frequently tested uni-modal interventions were: chloroquine/hydroxychloroquine (113 trials with 199,841 participants); convalescent plasma (64 trials with 11,840 participants); stem cells (51 trials with 3,370 participants); tocilizumab (19 trials with 4,139 participants) and favipiravir (19 trials with 3,210 participants).

### Methodological characteristics

Recruitment was registered as ongoing in 708 (54%) trial registry entries and completed in 110 (8%) (Table 1), of which 97 (88%) were from Iran. Blinded setups were planned in 607 (47%)



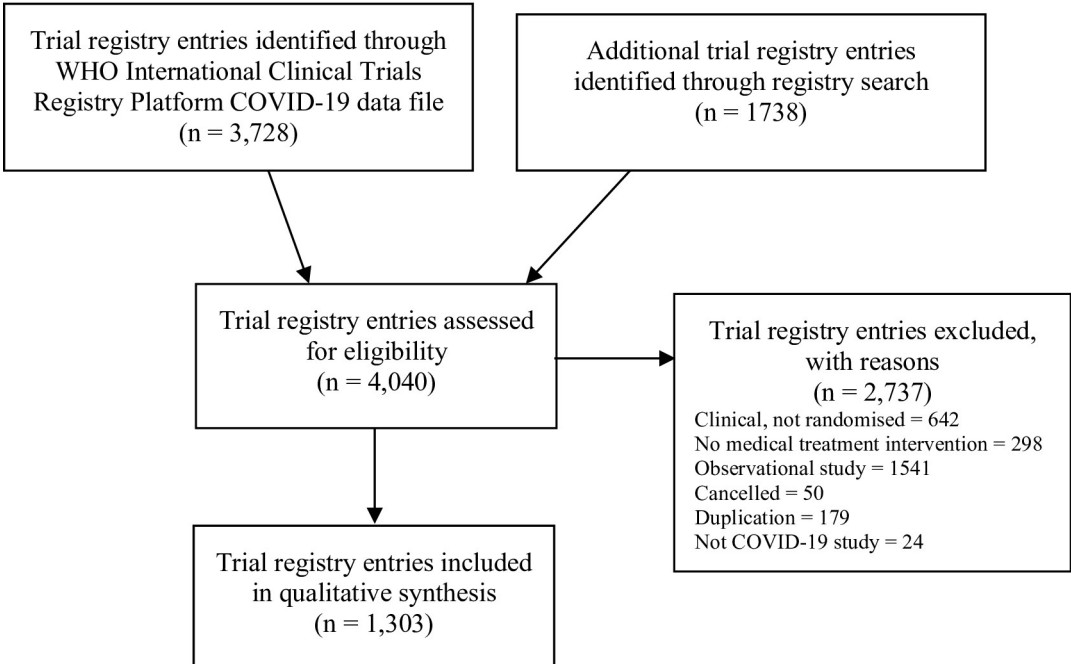

**Fig 1. PRISMA flowchart of trial registry entries for trials investigating COVID-19 treatment.** The last search was conducted June 23, 2020.

trial registry entries (Table 2). A total of 458 (35%) trial registry entries were categorized as multicenter investigations (Table 2). The planned sample size was >200 participants in 515 (40%) entries and the total number of participants planned for enrolment was 611,364 (Table 3). Mortality was planned to be assessed as a primary or secondary outcome in 748 (57%) trial registry entries with a total of 317,099 participants (Table 3). Participants were included regardless of sex in 1296 (99%) entries. Participants under 18 years of age were included in 95 (7%) entries.

## Geographical characteristics

The five countries with the highest number of registered trial registry entries were The United States (241 trials) accounting for 109,102 (18%) participants, China (239 trials) accounting for 64,707 (10%) participants, Iran (193 trials) accounting for 21,488 (4%) participants, Spain (80 trials) accounting for 28,625 (5%) participants and France (69 trials) accounting for 30,791 (5%) participants.

## Discussion

From January 23 to June 23, 2020 a total of 1,303 randomized clinical trials were registered on the 33 trial registries searched, investigating 381 therapeutics or adjunct therapies in treatment of COVID-19. The five most frequent investigational categories were immune modulating drugs, unconventional medicine, antimalarial drugs, antiviral drugs and respiratory adjuncts.

**Table 1. Characteristics of therapeutic groups.**

| Trial intervention | Trial registry entries | Recruitment ongoing | Recruitment completed | Disease specification | | | | Continents | | | | |
| | | | | Prevention | Diagnosed/ suspected COVID | Mild/ moderate symptoms | Hospitalized/ severe/ ICU | Asia | Europe | North America | South America | Africa |
| | n | n (%) | n (%) | n (%) | n (%) | n (%) | n (%) | n (%) | n (%) | n (%) | n (%) | n (%) |
| **All trials** | 1,303 | 708 (54%) | 110 (8%) | 160 (12%) | 275 (21%) | 203 (16%) | 665 (51%) | 556 (43%) | 313 (24%) | 290 (22%) | 54 (4%) | 48 (4%) |
| Unconventional medicine | 167 | 76 (46%) | 30 (18%) | 13 (8%) | 90 (54%) | 25 (15%) | 39 (23%) | 158 (95%) | 2 (1%) | 1 (1%) | 2 (1%) | 3 (2%) |
| Antimalarial drugs | 118 | 64 (54%) | 7 (6%) | 61 (52%) | 21 (18%) | 17 (14%) | 19 (16%) | 43 (36%) | 26 (22%) | 32 (27%) | 6 (5%) | 6 (5%) |
| Antiviral drugs | 100 | 51 (51%) | 13 (13%) | 9 (9%) | 21 (21%) | 27 (27%) | 43 (43%) | 56 (56%) | 10 (10%) | 14 (14%) | 8 (8%) | 8 (8%) |
| Antimalarial and antibiotics/antiviral drugs in comparison or combination | 46 | 23 (50%) | 3 (7%) | 6 (13%) | 4 (9%) | 10 (22%) | 26 (57%) | 15 (33%) | 13 (28%) | 11 (24%) | 3 (7%) | 3 (7%) |
| Antibodies | 82 | 50 (61%) | 7 (9%) | 2 (2%) | 5 (6%) | 8 (10%) | 67 (82%) | 28 (34%) | 13 (16%) | 29 (35%) | 9 (11%) | 2 (2%) |
| Anti-IL-6 | 33 | 26 (79%) | - | - | 2 (6%) | 5 (15%) | 26 (79%) | 3 (9%) | 16 (48%) | 10 (30%) | 1 (3%) | 1 (3%) |
| Other immunotherapeutic drugs | 151 | 93 (62%) | 4 (3%) | 5 (3%) | 22 (15%) | 18 (12%) | 106 (70%) | 43 (28%) | 46 (30%) | 49 (32%) | 2 (1%) | 2 (1%) |
| Respiratory adjuncts (inhaled, ventilator-related and gas-therapy) | 78 | 43 (55%) | 4 (5%) | 1 (1%) | 11 (14%) | 11 (14%) | 55 (71%) | 20 (26%) | 27 (35%) | 30 (38%) | - | - |
| Dietary supplements | 63 | 26 (41%) | 12 (19%) | 7 (11%) | 15 (24%) | 11 (17%) | 30 (48%) | 32 (51%) | 13 (21%) | 9 (14%) | 2 (3%) | 5 (8%) |
| Stem cells | 51 | 24 (47%) | 3 (6%) | 1 (2%) | 7 (14%) | 2 (4%) | 41 (80%) | 25 (49%) | 10 (20%) | 10 (30%) | 1 (3%) | 1 (2%) |
| Vaccine | 49 | 27 (55%) | - | 44 (90%) | 3 (6%) | 1 (2%) | 1 (2%) | 15 (31%) | 15 (31%) | 7 (14%) | 2 (4%) | 5 (10%) |
| Anti-inflammatory drugs | 45 | 31 (69%) | 4 (9%) | - | 14 (31%) | 6 (13%) | 25 (56%) | 20 (44%) | 14 (31%) | 10 (22%) | - | - |
| Glucocorticoids | 34 | 18 (53%) | 5 (15%) | - | 5 (15%) | 8 (24%) | 21 (62%) | 12 (35%) | 15 (44%) | 4 (12%) | 3 (9%) | - |
| Anticoagulants | 32 | 20 (63%) | - | - | 3 (9%) | 5 (16%) | 24 (75%) | 5 (16%) | 11 (34%) | 11 (34%) | 3 (9%) | - |
| Antihypertensives | 26 | 16 (62%) | 1 (4%) | 1 (4%) | 6 (23%) | 3 (12%) | 16 (62%) | 3 (12%) | 9 (35%) | 8 (31%) | 2 (8%) | 2 (8%) |
| Antibiotics/antifungal drugs | 27 | 13 (48%) | 4 (15%) | 1 (4%) | 9 (33%) | 6 (22%) | 11 (41%) | 8 (30%) | 10 (37%) | 7 (26%) | 1 (4%) | 1 (4%) |
| Other drug combinations | 79 | 46 (58%) | 5 (6%) | 3 (4%) | 17 (22%) | 18 (23%) | 41 (52%) | 26 (33%) | 32 (41%) | 9 (11%) | 3 (4%) | 7 (9%) |
| Other drugs | 122 | 61 (50%) | 8 (7%) | 6 (5%) | 20 (16%) | 22 (18%) | 74 (61%) | 44 (36%) | 31 (25%) | 39 (32%) | 5 (4%) | 2 (2%) |

**Table 2. Characteristics of therapeutic groups.**

| Trial intervention | Trial registry entries | Clinical research phase | | | | No. of blinded parties | | | | | Multicenter |
|---|---|---|---|---|---|---|---|---|---|---|---|
| | | 1 | 2 | 3 | 4 | None (open label) | 1 | 2 | 3 | 4 | |
| | n | n (%) | n (%) | n (%) | n (%) | n (%) | n (%) | n (%) | n (%) | n (%) | n (%) |
| **All trials** | 1,303 | 124 (10%) | 407 (31%) | 423 (32%) | 89 (7%) | 601 (46%) | 117 (9%) | 230 (18%) | 98 (8%) | 162 (12%) | 458 (35%) |
| **Unconventional medicine** | 167 | 25 (15%) | 17 (10%) | 41 (25%) | 4 (2%) | 76 (46%) | 9 (5%) | 33 (20%) | 11 (7%) | 5 (3%) | 41 (25%) |
| **Antimalarial drugs** | 118 | 17 (14%) | 20 (17%) | 55 (47%) | 16 (14%) | 43 (36%) | 9 (8%) | 24 (20%) | 10 (8%) | 24 (20%) | 37 (31%) |
| **Antiviral drugs** | 100 | 8 (8%) | 23 (23%) | 46 (46%) | 5 (5%) | 44 (44%) | 10 (10%) | 25 (25%) | 6 (6%) | 8 (8%) | 40 (40%) |
| **Antimalarial and antibiotics/antiviral drugs in comparison or combination** | 46 | 4 (9%) | 9 (20%) | 26 (57%) | 3 (7%) | 28 (61%) | 2 (4%) | 5 (11%) | 6 (13%) | 5 (11%) | 21 (46%) |
| **Antibodies** | 82 | 5 (6%) | 40 (49%) | 26 (32%) | - | 41 (50%) | 3 (4%) | 20 (24%) | 7 (9%) | 10 (12%) | 31 (38%) |
| **Anti-IL-6** | 33 | 1 (3%) | 19 (58%) | 11 (33%) | 2 (6%) | 17 (52%) | - | 9 (27%) | 1 (3%) | 5 (15%) | 15 (45%) |
| **Other immunotherapeutic drugs** | 151 | 17 (11%) | 74 (49%) | 42 (28%) | 8 (5%) | 69 (46%) | 9 (6%) | 25 (17%) | 11 (7%) | 29 (19%) | 76 (50%) |
| **Respiratory adjuncts (inhaled, ventilator-related and gas-therapy)** | 78 | 3 (4%) | 18 (23%) | 13 (17%) | 1 (1%) | 38 (49%) | 19 (24%) | 8 (10%) | 3 (4%) | 5 (6%) | 23 (29%) |
| **Dietary supplements** | 63 | 5 (8%) | 15 (24%) | 21 (33%) | 4 (6%) | 25 (40%) | 5 (8%) | 12 (19%) | 6 (10%) | 12 (19%) | 12 (19%) |
| **Stem cells** | 51 | 6 (12%) | 28 (55%) | 6 (12%) | - | 17 (33%) | 4 (8%) | 10 (20%) | 5 (10%) | 10 (20%) | 9 (18%) |
| **Vaccine** | 49 | 7 (14%) | 16 (33%) | 18 (37%) | 5 (10%) | 3 (6%) | 11 (22%) | 14 (29%) | 3 (6%) | 16 (33%) | 29 (59%) |
| **Anti-inflammatory drugs** | 45 | 5 (11%) | 17 (38%) | 17 (38%) | 3 (7%) | 21 (47%) | 6 (13%) | 8 (18%) | 3 (7%) | 4 (9%) | 14 (31%) |
| **Glucocorticoids** | 34 | 2 (6%) | 9 (26%) | 14 (41%) | 5 (15%) | 19 (56%) | 4 (12%) | 4 (12%) | 2 (6%) | 3 (9%) | 13 (38%) |
| **Anticoagulants** | 32 | - | 8 (25%) | 14 (44%) | 7 (22%) | 22 (69%) | 6 (19%) | 1 (3%) | 1 (3%) | 2 (6%) | 18 (56%) |
| **Antihypertensives** | 26 | 4 (15%) | 12 (46%) | 4 (15%) | 4 (15%) | 12 (46%) | 2 (8%) | 3 (12%) | 4 (15%) | 4 (15%) | 8 (31%) |
| **Antibiotics/antifungal drugs** | 27 | - | 10 (37%) | 14 (52%) | 2 (7%) | 15 (56%) | 1 (4%) | 4 (15%) | 3 (11%) | 4 (15%) | 9 (33%) |
| **Other drug combinations** | 79 | 6 (8%) | 23 (29%) | 33 (42%) | 11 (14%) | 55 (70%) | 7 (9%) | 6 (8%) | 4 (5%) | 3 (4%) | 30 (38%) |
| **Other drugs** | 122 | 9 (7%) | 49 (40%) | 22 (18%) | 9 (7%) | 56 (46%) | 10 (8%) | 19 (16%) | 12 (10%) | 13 (11%) | 32 (26%) |

Target sizes were above 200 participants in 40%, blinding was used in 47% and mortality was registered as an outcome in 57% of trial registry entries.

We found a steep increase in trial registry entries on RCTs, currently focusing on immune modulating-, antimalarial- and antiviral drugs. Ongoing trials initiated before the pandemic and now including COVID-19 patients are not described in this review, but can provide important information on treatment modalities and should be continued [15].

Early trial registry entries from Asia generally had smaller trial sizes. Small trial sizes can lead to overestimated intervention effects and underestimated harms [16]. Entries in the Iranian Registry of Clinical Trials (IRCT) accounted for 15% of all registered RCTs, but only for

**Table 3. Characteristics of therapeutic groups.**

| Trial intervention | Trial registry entries | Sample size | | | | Total participants | Mortality outcome | | |
|---|---|---|---|---|---|---|---|---|---|
| | | < 50 | 50–199 | 200–999 | ≥1000 | | Primary | Secondary | Participants assessed for mortality |
| | n | n (%) | n (%) | n (%) | n (%) | n | n (%) | n (%) | n (%) |
| **All trials** | 1,303 | 224 (17%) | 564 (43%) | 392 (30%) | 123 (9%) | 611,364 | 278 (21%) | 470 (36%) | 317,099 (52%) |
| **Unconventional medicine** | 167 | 20 (12%) | 110 (66%) | 33 (20%) | 4 (2%) | 47,232 | 12 (7%) | 26 (16%) | 6,018 (13%) |
| **Antimalarial drugs** | 118 | 6 (5%) | 27 (23%) | 54 (46%) | 31 (26%) | 200,583 | 16 (14%) | 41 (35%) | 51,951 (26%) |
| **Antiviral drugs** | 100 | 16 (16%) | 57 (57%) | 23 (23%) | 4 (4%) | 19,554 | 8 (8%) | 35 (35%) | 7,303 (37%) |
| **Antimalarial and antibiotics/antiviral drugs in comparison or combination** | 46 | 2 (4%) | 13 (28%) | 18 (39%) | 13 (28%) | 36,069 | 18 (39%) | 14 (30%) | 32,062 (89%) |
| **Antibodies** | 82 | 17 (21%) | 43 (52%) | 19 (23%) | 3 (4%) | 15,006 | 30 (37%) | 33 (40%) | 12,794 (85%) |
| **Anti-IL-6** | 33 | 5 (15%) | 12 (36%) | 16 (48%) | - | 6,485 | 9 (27%) | 22 (67%) | 6,215 (96%) |
| **Other immunotherapeutic drugs** | 151 | 39 (26%) | 68 (45%) | 42 (28%) | 2 (1%) | 25,122 | 37 (25%) | 70 (46%) | 19,155 (76%) |
| **Respiratory adjuncts (inhaled, ventilator-related and gas-therapy)** | 78 | 17 (22%) | 28 (36%) | 32 (41%) | 1 (1%) | 18,480 | 17 (22%) | 33 (42%) | 15,082 (82%) |
| **Dietary supplements** | 63 | 12 (19%) | 29 (46%) | 15 (24%) | 7 (11%) | 18,337 | 10 (16%) | 27 (43%) | 13,558 (74%) |
| **Stem cells** | 51 | 28 (55%) | 20 (39%) | 3 (6%) | - | 3,370 | 16 (31%) | 14 (27%) | 2,124 (63%) |
| **Vaccine** | 49 | 1 (2%) | 7 (14%) | 16 (33%) | 25 (51%) | 76,922 | 3 (6%) | 15 (31%) | 32,920 (43%) |
| **Anti-inflammatory drugs** | 45 | 9 (20%) | 19 (42%) | 12 (27%) | 5 (11%) | 17,305 | 9 (20%) | 18 (40%) | 15,257 (88%) |
| **Glucocorticoids** | 34 | 3 (9%) | 12 (35%) | 18 (53%) | 1 (3%) | 8,503 | 10 (29%) | 16 (47%) | 6,225 (73%) |
| **Anticoagulants** | 32 | 2 (6%) | 12 (38%) | 14 (44%) | 4 (13%) | 15,348 | 13 (41%) | 13 (41%) | 14,090 (92%) |
| **Antihypertensives** | 26 | 3 (12%) | 7 (27%) | 14 (54%) | 2 (8%) | 18,710 | 12 (46%) | 9 (35%) | 17,474 (93%) |
| **Antibiotics/antifungal drugs** | 27 | 6 (22%) | 8 (30%) | 12 (44%) | 1 (4%) | 8,199 | 4 (15%) | 11 (41%) | 5,793 (71%) |
| **Other drug combinations** | 79 | 9 (11%) | 29 (37%) | 22 (28%) | 19 (24%) | 56,271 | 20 (25%) | 34 (43%) | 44,633 (79%) |
| **Other drugs** | 122 | 29 (24%) | 63 (52%) | 29 (24%) | 1 (1%) | 19,868 | 34 (28%) | 39 (32%) | 14,445 (73%) |

4% of all planned participants. Maybe because of the generally smaller trial sizes in entries from the IRCT, this registry contributed with 85% of all trials that had completed recruitment per June 23.

Though there is no firm evidence that lack of blinding affects estimates of mortality [17], we consider blinding as an important factor in COVID-19 trial designs. Especially as 79% of trial registry entries had other primary outcomes than mortality and many were preventive with subjective and patient-reported outcomes such as self-assessed symptom severity where bias, including lack of blinding, that can exaggerate intervention effects [18].

**Table 4. Characteristics of specific interventions reported in seven or more trial registry entries.**

A

| Trial intervention | Trial registry entries | Recruitment ongoing | Recruitment completed | Disease specification | | | | Continents | | | | | Multicenter |
|---|---|---|---|---|---|---|---|---|---|---|---|---|---|
| | | | | Prevention | Diagnosed/ suspected COVID | Mild/ moderate symptoms | Hospitalized/ severe/ ICU | Asia | Europe | North America | South America | Africa | |
| | n | n (%) | n (%) | n (%) | n (%) | n (%) | n (%) | n (%) | n (%) | n (%) | n (%) | n (%) | n (%) |
| Chloroquine | 113 | 62 (55%) | 7 (6%) | 60 (53%) | 21 (19%) | 15 (13%) | 17 (15%) | 42 (37%) | 24 (21%) | 32 (28%) | 6 (5%) | 4 (4%) | 36 (32%) |
| Convalescent Plasma | 64 | 38 (59%) | 5 (8%) | 1 (2%) | 5 (8%) | 5 (8%) | 53 (83%) | 23 (36%) | 11 (17%) | 19 (30%) | 9 (14%) | 2 (3%) | 22 (34%) |
| Stem cells | 51 | 24 (47%) | 3 (6%) | 1 (2%) | 7 (14%) | 2 (4%) | 41 (80%) | 25 (49%) | 10 (20%) | 10 (20%) | 2 (4%) | 1 (2%) | 9 (18%) |
| Chloroquine + other | 30 | 17 (57%) | 4 (13%) | 4 (13%) | 5 (17%) | 4 (13%) | 17 (57%) | 12 (40%) | 14 (47%) | 2 (7%) | 0 | 0 | 17 (57%) |
| Chloroquine / Azithromycin | 22 | 9 (41%) | 0 | 2 (9%) | 2 (9%) | 5 (23%) | 13 (59%) | 4 (18%) | 6 (27%) | 8 (36%) | 3 (14%) | 1 (5%) | 7 (32%) |
| Tocilizumab | 19 | 15 (79%) | 0 | 0 | 1 (5%) | 3 (16%) | 15 (79%) | 3 (16%) | 9 (47%) | 4 (21%) | 1 (5%) | 0 | 11 (58%) |
| Favipiravir | 19 | 10 (53%) | 2 (11%) | 0 | 2 (11%) | 10 (53%) | 7 (37%) | 13 (68%) | 2 (11%) | 2 (11%) | 0 | 2 (11%) | 7 (37%) |
| BCG vaccine | 17 | 12 (71%) | 0 | 17 (100%) | 0 | 0 | 0 | 3 (18%) | 8 (47%) | 1 (6%) | 2 (12%) | 2 (12%) | 14 (82%) |
| Ivermectin | 17 | 9 (53%) | 2 (12%) | 2 (12%) | 2 (12%) | 7 (41%) | 6 (35%) | 7 (41%) | 4 (24%) | 0 | 4 (24%) | 2 (12%) | 5 (29%) |
| Prone position | 16 | 13 (81%) | 0 | 0 | 0 | 3 (19%) | 13 (81%) | 1 (6%) | 6 (38%) | 9 (56%) | 0 | 0 | 7 (44%) |
| Colchicine | 16 | 15 (94%) | 1 (6%) | 0 | 5 (31%) | 4 (25%) | 7 (44%) | 4 (25%) | 7 (44%) | 4 (25%) | 0 | 0 | 7 (44%) |
| Azithromycin | 15 | 8 (53%) | 2 (13%) | 0 | 5 (33%) | 3 (20%) | 7 (47%) | 4 (27%) | 7 (47%) | 2 (13%) | 1 (7%) | 1 (7%) | 6 (40%) |
| Enoxaparin | 11 | 7 (64%) | 0 | 0 | 0 | 4 (36%) | 7 (64%) | 2 (18%) | 4 (36%) | 3 (27%) | 2 (18%) | 0 | 5 (45%) |
| Vitamin C | 10 | 2 (20%) | 4 (40%) | 0 | 1 (10%) | 1 (10%) | 8 (80%) | 6 (60%) | 0 | 3 (30%) | 0 | 0 | 1 (10%) |
| Nitric Oxide | 10 | 5 (50%) | 0 | 1 (10%) | 1 (10%) | 3 (30%) | 5 (50%) | 0 | 0 | 10 (100%) | 0 | 0 | 2 (20%) |
| Ozone autohemotherapy | 9 | 4 (44%) | 2 (22%) | 0 | 2 (22%) | 1 (11%) | 6 (67%) | 4 (44%) | 4 (44%) | 1 (11%) | 0 | 0 | 2 (22%) |
| INF-Beta | 9 | 5 (56%) | 2 (22%) | 0 | 4 (44%) | 1 (11%) | 4 (44%) | 8 (89%) | 1 (11%) | 0 | 0 | 0 | 2 (22%) |
| Glucocorticoid | 9 | 5 (56%) | 1 (11%) | 0 | 1 (11%) | 1 (11%) | 7 (78%) | 5 (56%) | 2 (22%) | 0 | 2 (22%) | 0 | 3 (33%) |
| Vitamin D | 8 | 5 (63%) | 0 | 0 | 2 (25%) | 2 (25%) | 4 (50%) | 1 (13%) | 5 (63%) | 1 (13%) | 1 (13%) | 0 | 3 (38%) |
| Sarilumab | 8 | 6 (75%) | 0 | 0 | 2 (25%) | 2 (25%) | 5 (63%) | 0 | 6 (75%) | 2 (25%) | 0 | 0 | 2 (25%) |
| Vitamin A | 7 | 2 (29%) | 2 (29%) | 0 | 2 (29%) | 1 (14%) | 4 (57%) | 6 (86%) | 0 | 0 | 0 | 1 (14%) | 1 (14%) |
| Heparin | 7 | 5 (71%) | 0 | 0 | 0 | 0 | 7 (100%) | 0 | 1 (14%) | 5 (71%) | 0 | 0 | 4 (57%) |

(*Continued*)

**Table 4.** (Continued)

**B**

| Trial intervention | Trial registry entries | No. of blinded parties | | | | | Sample size | | | | Total participants | Mortality outcome | | |
|---|---|---|---|---|---|---|---|---|---|---|---|---|---|---|
| | | None (open label) | 1 | 2 | 3 | 4 | <50 | 50–199 | 200–999 | ≥1000 | | Primary | Secondary | Participants assessed for mortality |
| | n | n (%) | n (%) | n (%) | n (%) | n (%) | n (%) | n (%) | n (%) | n (%) | n | n (%) | n (%) | n (%) |
| **Chloroquine** | 113 | 41 (36%) | 9 (8%) | 23 (20%) | 9 (8%) | 24 (21%) | 5 (4%) | 25 (22%) | 52 (46%) | 31 (27%) | 199,841 | 16 (14%) | 39 (35%) | 51,829 (26%) |
| **Convalescent Plasma** | 64 | 37 (58%) | 3 (5%) | 12 (19%) | 7 (11%) | 4 (6%) | 12 (19%) | 33 (52%) | 17 (27%) | 2 (3%) | 11,840 | 27 (42%) | 25 (39%) | 10,331 (87%) |
| **Stem cells** | 51 | 17 (33%) | 4 (8%) | 10 (20%) | 5 (10%) | 10 (20%) | 28 (55%) | 20 (39%) | 3 (6%) | 0 | 3,370 | 16 (31%) | 14 (27%) | 2,124 (63%) |
| **Chloroquine + other** | 30 | 21 (70%) | 0 | 3 (10%) | 2 (7%) | 3 (10%) | 0 | 9 (30%) | 8 (27%) | 13 (43%) | 46,944 | 11 (37%) | 12 (40%) | 39,314 (84%) |
| **Chloroquine / Azithromycin** | 22 | 13 (59%) | 1 (5%) | 2 (9%) | 2 (9%) | 4 (18%) | 2 (9%) | 5 (23%) | 11 (50%) | 4 (18%) | 12,186 | 5 (23%) | 8 (36%) | 9,960 (82%) |
| **Tocilizumab** | 19 | 10 (53%) | 0 | 7 (37%) | 0 | 1 (5%) | 1 (5%) | 9 (47%) | 9 (47%) | 0 | 4,139 | 7 (37%) | 12 (63%) | 4,139 (100%) |
| **Favipiravir** | 19 | 14 (74%) | 0 | 4 (21%) | 0 | 0 | 2 (11%) | 12 (63%) | 4 (21%) | 1 (5%) | 3,210 | 0 | 8 (42%) | 1,232 (38%) |
| **BCG vaccine** | 17 | 2 (12%) | 4 (24v) | 3 (18%) | 0 | 8 (47%) | 0 | 0 | 4 (24%) | 13 (76%) | 27,262 | 1 (6%) | 10 (59%) | 15,616 (57%) |
| **Ivermectin** | 17 | 5 (29%) | 1 (6%) | 7 (41%) | 1 (6%) | 3 (18%) | 3 (18%) | 12 (71%) | 2 (12%) | 0 | 1,752 | 1 (6%) | 1 (6%) | 326 (19%) |
| **Prone position** | 16 | 12 (75%) | 3 (19%) | 1 (6%) | 0 | 0 | 0 | 5 (31%) | 11 (69%) | 0 | 4,228 | 8 (50%) | 5 (31%) | 3,612 (85%) |
| **Colchicine** | 16 | 9 (56%) | 2 (13%) | 3 (19%) | 1 (6%) | 1 (6%) | 1 (6%) | 8 (50%) | 3 (19%) | 4 (25%) | 12,268 | 4 (25%) | 6 (38%) | 11,624 (95%) |
| **Azithromycin** | 15 | 10 (67%) | 1 (7%) | 1 (7%) | 1 (7%) | 2 (13%) | 1 (7%) | 6 (40%) | 7 (47%) | 1 (7%) | 5,591 | 3 (20%) | 8 (53%) | 4,893 (88%) |
| **Enoxaparin** | 11 | 10 (91%) | 1 (9%) | 0 | 0 | 0 | 1 (9%) | 4 (36%) | 3 (27%) | 3 (27%) | 6,134 | 5 (45%) | 4 (36%) | 6,014 (98%) |
| **Vitamin C** | 10 | 2 (20%) | 1 (10%) | 3 (30%) | 1 (10%) | 3 (30%) | 2 (20%) | 5 (50%) | 3 (30%) | 0 | 1,706 | 1 (10%) | 6 (60%) | 1,490 (87%) |
| **Nitric Oxide** | 10 | 5 (50%) | 1 (10%) | 1 (10%) | 1 (10%) | 2 (20%) | 4 (40%) | 0 | 6 (60%) | 0 | 2,082 | 0 | 7 (70%) | 1,292 (62%) |
| **Ozone autohemotherapy** | 9 | 3 (33%) | 4 (44%) | 1 (11%) | 0 | 0 | 2 (22%) | 6 (67%) | 1 (11%) | 0 | 872 | 1 (11%) | 3 (33%) | 470 (54%) |
| **INF-Beta** | 9 | 4 (44%) | 0 | 2 (22%) | 1 (11%) | 1 (11%) | 3 (33%) | 3 (33%) | 3 (33%) | 0 | 1,348 | 1 (11%) | 4 (44%) | 630 (47%) |
| **Glucocorticoid** | 9 | 5 (56%) | 1 (11%) | 0 | 0 | 1 (11%) | 1 (11%) | 4 (44%) | 4 (44%) | 0 | 1,542 | 3 (33%) | 4 (44%) | 1,342 (87%) |
| **Vitamin D** | 8 | 3 (38%) | 0 | 2 (25%) | 0 | 3 (38%) | 0 | 3 (38%) | 2 (25%) | 3 (38%) | 4,487 | 4 (50%) | 3 (38%) | 4,423 (99%) |
| **Sarilumab** | 8 | 6 (75%) | 0 | 0 | 0 | 2 (25%) | 1 (13%) | 2 (25%) | 5 (63%) | 0 | 1,726 | 2 (25%) | 5 (63%) | 1,486 (86%) |
| **Vitamin A** | 7 | 5 (71%) | 0 | 1 (14%) | 1 (14%) | 0 | 2 (29%) | 3 (43%) | 2 (29%) | 0 | 1,089 | 0 | 2 (29%) | 405 (37%) |
| **Heparin** | 7 | 2 (29%) | 2 (29%) | 0 | 1 (14%) | 2 (29%) | 0 | 3 (43%) | 3 (43%) | 1 (14%) | 4,454 | 3 (43%) | 3 (43%) | 4,354 (98%) |

The number of trial registry entries from Europe and the US has been rapidly increasing and are generally of larger sample sizes. Further, international multicenter trials are planned to include huge numbers of participants–currently eight with above 10,000 participants. Collaboration to compile evidence in meta-analyses and network meta-analyses in order to achieve higher levels of evidence has been initiated, such as The Living Mapping and Living Systematic Review of COVID-19 Studies [19, 20]. These websites can be very useful in finding trial registry entries for specific pharmacological treatment categories and dynamic changes in activity. The current review builds upon these works by providing a per June 23 updated snapshot overview, with inclusion of trial registry entries from an additional 22 trial registries and removal of duplicates, on research activity for both treatment categories and specific drugs with details on mortality assessment and blinding.

COVID-19 mortality rates and overcrowded intensive care units underline the importance of identifying therapeutics and adjunct therapies that can reduce mortality and shorten hospital admissions, and trials should assess these outcomes whenever appropriate. Accordingly, a core outcome set qualified by a Delphi process has been developed for clinical trials on COVID-19, which included all-cause mortality for ordinary, severe and critical COVID-19 infection [21]. RCTs on COVID-19 treatment have been criticized for not being sufficiently powered to assess mortality [22], which seem justifiable as a large part of trial registry entries with small trial sizes used mortality as an outcome. Mortality was an outcome in 57% of the included trial registry entries, which is encouraging although improvement is possible. Mortality as a primary outcome requires larger sample sizes, which can affect the conductibility of some trials, but even inclusion of mortality as a secondary or exploratory outcome can be important, as such data will add essential information for future meta-analyses. We endorse adherence to the core outcome set, as it will facilitate homogeneous outcome reporting and improve the quality of evidence in up-coming meta-analyses and subsequent evidence-based recommendations [21, 23].

Clinical research is time consuming and costly. In time of a pandemic, worldwide research collaboration is encouraged [24]. Several cases of such partnerships have been developed including the Adaptive COVID-19 Treatment Trial (ACTT) that investigate remdesivir for COVID-19 treatment in up to 75 study locations [25], and the active recruiting WHO initiated Solidarity trial collaboration that tests four different active treatments (remdesivir, chloroquine/hydroxy-chloroquine, lopinavir + ritonavir, or lopinavir + ritonavir + interferon beta-1a) versus local standard of care [26]. According to the WHO homepage, July 1, 2020, 39 countries have approvals to begin recruiting and 5,500 participants have already been recruited in 21 countries [26].

## Strengths and limitations

The comprehensive and structured search in public registries, double screening for inclusion, transfer of trial registry entry data from original registers, independent parallel screening and extraction and assessment of blinding status are strengths of this study. Further, we updated our search, the status of recruitment and cancellations just prior to submission. We screened for registrations in multiple registers, but we may have missed some due to incomplete, late, or heterogeneous data registration in trial registries. Some included trial registry entries may subsequently be ethically denied, retracted, or fail to reach planned trial size. Altogether, this means that our results may overestimate the number of participants that will be randomized to each intervention and the number of trials that will reach publication.

This is a dynamic and rapidly developing research field and with the present study, we only provide a snapshot overview of trial registry entries of June 23, 2020. We however, believe that

the overview of investigated interventions and quantification of several methodological parameters, not provided elsewhere, have merits. More adaptable inventories for finding trial registry entries of specific interventions are available elsewhere [11, 12, 27]. We reviewed trial registry entries and not actual trial protocols, mainly because these were not always available. Evaluation of trial protocols could have provided more detailed information on trial methodology and outcomes and we endorse recent suggestions of public access to all trial protocols concomitant with trial registration [28]. Categorisation of interventions is to some extent arbitrary as some drugs fit into more categories. We chose to categorize trials that tested multiple interventions in factorial design or as multimodal regimens into one heterogenic group to maintain overview.

## Perspective

An encouraging amount of research has been launched to fight SARS-CoV-2. We expect trialists can use this review to get an overview of the ongoing research related to COVID-19 treatment which could help prioritize and design future clinical trials.

## Conclusion

An extraordinary number of randomized clinical trials investigating COVID-19 management have been initiated with a multitude of medical preventive, adjunctive and treatment modalities. Blinding was used in only 47% of trial registry entries, which may influence future reported treatment effects. Fifty-seven percent of all trials will assess mortality as an outcome facilitating future meta-analyses. Large multicenter and international collaborative trials are being prioritized and many are currently recruiting.

## Supporting information

**S1 Appendix. Trial registration sites.**
(PDF)

**S2 Appendix. Excluded trial registry entries.**
(PDF)

**S3 Appendix. Included trial trial registry entries.**
(PDF)

**S4 Appendix. Database.**
(CSV)

**S1 Checklist. PRISMA 2009 checklist.**
(DOC)

## Author Contributions

**Conceptualization:** Anders Peder Højer Karlsen, Sebastian Wiberg, Ole Mathiesen.

**Data curation:** Anders Peder Højer Karlsen, Sebastian Wiberg, Jens Laigaard, Casper Pedersen, Kim Zillo Rokamp, Ole Mathiesen.

**Formal analysis:** Anders Peder Højer Karlsen, Sebastian Wiberg, Jens Laigaard, Casper Pedersen, Kim Zillo Rokamp.

**Investigation:** Anders Peder Højer Karlsen, Sebastian Wiberg, Jens Laigaard, Casper Pedersen, Kim Zillo Rokamp.

**Methodology:** Anders Peder Højer Karlsen, Sebastian Wiberg, Ole Mathiesen.

**Project administration:** Anders Peder Højer Karlsen.

**Resources:** Anders Peder Højer Karlsen, Jens Laigaard, Casper Pedersen.

**Supervision:** Anders Peder Højer Karlsen, Kim Zillo Rokamp, Ole Mathiesen.

**Validation:** Anders Peder Højer Karlsen, Sebastian Wiberg, Kim Zillo Rokamp, Ole Mathiesen.

**Visualization:** Anders Peder Højer Karlsen, Sebastian Wiberg, Jens Laigaard, Casper Pedersen.

**Writing – original draft:** Anders Peder Højer Karlsen, Ole Mathiesen.

**Writing – review & editing:** Anders Peder Højer Karlsen, Sebastian Wiberg, Jens Laigaard, Casper Pedersen, Kim Zillo Rokamp, Ole Mathiesen.

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
