## [Decision Letter · Decision Letter 0]

9 Jul 2020

PONE-D-20-13626

A systematic review of submitted protocols for randomized clinical trials investigating COVID-19 medical prevention and treatment

PLOS ONE

Dear Dr. Karlsen,

Thank you for submitting your manuscript to PLOS ONE. After careful consideration, we feel that it has merit but does not fully meet PLOS ONE’s publication criteria as it currently stands. Therefore, we invite you to submit a revised version of the manuscript that addresses the points raised during the review process.

A key issue noted by both reviewers is the incorrect reference to trial protocols when the meaning is actually trial registrations. Both reviewers also have additional comments which should be responded to.

We look forward to receiving your revised manuscript.

Kind regards,

Lisa Susan Wieland

Academic Editor

PLOS ONE

Journal Requirements:

2. Please confirm that you have included all items recommended in the PRISMA checklist including the full electronic boolean search strategy used to identify studies with all search terms and limits for at least one database.

Please attach this as supplementary file.

3. In your discussion, please explain how your work builds upon similar previously published research, including

(1) https://covid-nma.com/ and

(2) https://www.iddo.org/research-themes/covid-19/live-systematic-clinical-trial-review.

Reviewers' comments:

Reviewer's Responses to Questions

**Comments to the Author**

1. Is the manuscript technically sound, and do the data support the conclusions?

Reviewer #1: Yes

Reviewer #2: Yes

2. Has the statistical analysis been performed appropriately and rigorously? 

Reviewer #1: N/A

Reviewer #2: Yes

3. Have the authors made all data underlying the findings in their manuscript fully available?

Reviewer #1: Yes

Reviewer #2: Yes

4. Is the manuscript presented in an intelligible fashion and written in standard English?

Reviewer #1: Yes

Reviewer #2: Yes

5. Review Comments to the Author

Reviewer #1: Thank you for the opportunity to review this manuscript.

The manuscript presents a rigorous study examining which clinical studies had been registered in 32 different registries. The manuscript presents the planned size, planned blinding, and whether mortality was a planned outcome for the identified planned studies. The authors suggest that the results of the study can be used to plan future studies and set research priorities. Unfortunately, considering the rapidly evolving situation I am afraid the findings might already be outdated. Additionally, several regularly updated alternatives exist, e.g. Cochranes “Covid-19 Study Register” (https://covid-19.cochrane.org/?pn=1), EBM Datalabs “Covid-19 TrialsTracker” (http://covid19.trialstracker.net/), and COVID-Evidence (https://covid-evidence.org/database)

Thus, I am afraid the reviewed manuscript will not be able to contribute substantially to priority setting or planning of future research activities: I do, however, find the methods to be technically sound and that the data supports the conclusions. My comments to the manuscript are presented below:

Major issues:

Introduction:

• Several resources that track planned and completed trials are available online (including, but not limited to, the ones mentioned above). These should be mentioned in the introduction and it should be clarified how the presented study contributes.

• Throughout the manuscript, the authors refer to protocols. A clinical study protocol is a quite elaborate document, that is e.g. submitted to ethical committees or institutional review boards to get ethical approval. Protocols are sometimes, but not always, published, but would not necessarily be available from trial registries, so it is my impression that the authors are in fact referring to trial registry entries. If the authors have in fact obtained protocols for all included trials, they should explain how these were obtained. Otherwise, I suggest changing the wording from protocol to trial registry entry throughout the text.

Methods:

• It is not explained why blinding is considered of particular importance. I would probably expect blinding to be less important for relatively objective outcomes such as overall mortality (although blinding could be very important for cause-specific mortality). Perhaps the authors could elaborate on the choice of outcome.

o Additionally, in the results, the authors use the categories “Open label, single blind, double blind, triple blind, and quadruple blind”. It is not clear what these different categories describe, and research has shown that e.g. “double blind” is an ambiguous term[1,2]. The authors should explain how they operationalise blinding in the methods section.

• Regarding mortality as an outcome: Are the authors only looking at overall mortality or also cause-specific? This should be elaborated on.

Discussion:

• Perspective: as mentioned above, I am afraid that the results of the study may already be outdated and more up-to-date resources exist.

Minor issues:

Throughout the text the terms blinding, and masking are used interchangeably. If the authors believe these words are not describing the same this should be elaborated on, otherwise the text should be revised so only one term is used.

Abstract

• In conclusions the authors write: “a second wave of higher level evidence” – it is unclear what the first wave was and thus what “higher” is relative to. This is explained in the introduction but is not mentioned in the abstract. Thus, I suggest rephrasing.

Background:

• First paragraph: the authors write: “symptoms ranging from mild symptoms of upper airway infection to …” – I believe many reports state that a high proportion of people infected with SARS-CoV-2 are completely asymptomatic. Perhaps the authors could consider mentioning this.

• Second paragraph: the authors write: “… to ensure a second wave of high-quality evidence with proper assessment of safety and efficacy measures” – I would suggest avoiding the term “safety” as this term tends to underplay harms and convey the idea that drugs have no, or non-important, side effects. I would suggest using “proper assessment of benefits and harms” – however, I am aware that this is not universally agreed upon, so just a suggestion.

Methods:

• Data handling: The authors refer to “trial phase” – it is not immediately clear to me what they are referring to here – I assume it is the phases of clinical research, but some trials have multiple phases (e.g. double-blind and open-label), so perhaps the authors could elaborate a bit.

Results:

• First paragraph: The authors exclude a high proportion of identified studies; I would suggest mentioning the main reasons for exclusion, although these are also available from the PRISMA flowchart

• Table 1 and Table 2: I suggest indenting the different types of interventions under “All trials” to make the table more legible.

• Table 4a: The recruiting / completed variable is somewhat confusing. Completed could easily be interpreted as “trial completed” rather than “recruiting completed”.

Discussion:

• First paragraph: The first sentence is perhaps a bit too strongly worded, 770 RCTs were registered on the trial registries searched, there might be trials registered elsewhere.

• Second paragraph, second line: delete “from” in “ongoing trials initiated from before the pandemic”.

• Second paragraph, fourth line: Rephrase the following sentence “China published 30% of the protocols”. The trials were not registered by China but from China. Also again, these are not published protocols but registry entries.

• Perhaps mention that such initiatives exist, e.g. the Living Meta-analysis from Cochrane France and others: https://covid-nma.com/

• Strengths and limitations: It is not mentioned in “methods” that data was extracted in duplicate.

• Strengths and limitations: The authors use the term “quality parameters”. Firstly, it is unclear to me what this refers to, I suppose blinding but that is only one parameter. Secondly, I would suggest avoiding the term “quality” which is somewhat normative and just call it blinding status instead.

• Strengths and limitations: I am unsure why including “unconventional medicine” would be controversial, it is an important part of the narrative of the research being conducted.

Supplementary information:

Appendix 2 and appendix 3 are called “Excluded articles” and “included articles”, however the unit of analysis is not articles but registry entries.

 

References

1 Hróbjartsson A, Pildal J, Chan A-W, et al. Reporting on blinding in trial protocols and corresponding publications was often inadequate but rarely contradictory. J Clin Epidemiol 2009;62:967–73. doi:10.1016/j.jclinepi.2009.04.003

2 Devereaux PJ, Manns BJ, Ghali WA, et al. Physician Interpretations and Textbook Definitions of Blinding Terminology in Randomized Controlled Trials. JAMA 2001;285:2000–3. doi:10.1001/jama.285.15.2000

Reviewer #2: This manuscript describes the results of a systematic search for registered drug trials in the WHO trial registry that assess prevention or treatment for COVID-19. This is important to provide an overview of ongoing research, as well as its characteristics and quality. The authors assess kkey components of the trials, such as number of participants, outcome measures, blinding, status for recruiting, etc.

Overall, the methods and results are clearly presented and the authors use blinded data-extraction, a systematic search strategy, present a PRISMA flow-chart with reasons for exclusions, etc.

My main concern is that the authors have likely not, as stated in the manuscript, assessed the actual protocols for the trials. Trial registries generally do not included detailed protocols but a set of key pieces of information, which is a pity. Lots of important information is missing that would allow a more thorough assessment of the quality of the ongoing trials than just binding, e.g. by applying the Cochrane risk of bias tool and assess whether the randosmisation process was adequate and likely to produce comparable groups. Given that I am correct about this, it might be necessary to change the terminology of the manuscript and explain that it is an evaluation of clinical trial registry forms rather than protocols and perhaps even use this to push for requirements to publish the full protocol along with the trial registry forms.

6. PLOS authors have the option to publish the peer review history of their article (what does this mean?). If published, this will include your full peer review and any attached files.

Reviewer #1: **Yes: **Asger Sand Paludan-Müller

Reviewer #2: **Yes: **Karsten Juhl Jørgensen

---

## [Author Response · Author response to Decision Letter 0]

21 Jul 2020

15th of July 2020

Response to editor

Dear editor,

Thank you very much for considering our manuscript for publication and for your thorough comments. We have been able to address all comments/concerns, and hope you agree that the manuscript has improved considerably. We have inserted the reviewer comments below, and will hereby address their points one by one, with our response in blue font. In addition, a copy of the manuscript with “track changes” will be uploaded. Text changes in the manuscript are written in red font.

Kind regards,

On behalf of the authors:

Dr. Anders Karlsen, MD, PhD

General comment.

Thank you for stating that "Collaboration to compile evidence in meta-analyses and network meta-analyses in order to achieve higher levels of evidence has been initiated, such as The Living Mapping and Living Systematic Review of COVID-19 Studies [19-20].". Please elaborate on how your work builds upon these similar previously published works. Please provide this information in your discussion.

Author response: Thank you for this comment. These websites have evolved towards more detail since our first submission and now provides important information on COVID-19 research activity. Still, the overview per July 21 on https://www.iddo.org/tool/covid-19-clinical-trials-interactive-tool is limited to 567 prospective trials and https://covid-nma.com/dataviz/# does not provide information on blinding, mortality. Further, the covid-nma overview function is limited to treatment categories and not specific interventions as we report in table 4. Further, by checking registry sites and manually screening our database we found and excluded 126 duplicate registry entries with 95,386 participants specifically from the ICTRP. In the https://covid-nma.com/dataviz/# we find duplicates, such as EUCTR-2020-001200-42-DK/NCT04321096 and ISRCTN14326006/NCT04374942 which may lead to overestimation of research activity. We send an email notification regarding this issue, but chose not to comment in the manuscript as they may choose to remove duplicates.

We added to the discussion: 

”These websites can be very useful in finding trial registry entries for specific pharmacological treatment categories and dynamic changes in activity. The current review builds upon these works by providing a per June 23 updated snapshot overview, with inclusion of trial registry entries from an additional 22 trial registries and removal of duplicates, on research activity for both treatment categories and specific drugs with details on mortality assessment and blinding.” (page 13, line 27-31).

Further we added to the strength and limitation section: 

“This is a dynamic and rapidly developing research field and with the present study, we only provide a snapshot overview of trial registry entries of June 23, 2020. We however, believe that the overview of investigated interventions and quantification of several methodological parameters, not provided elsewhere, have merits. More adaptable inventories to find trial registry entries of specific interventions are available elsewhere [11, 12, 27].” (page 15, line 3-7).

Reviewer #1: Thank you for the opportunity to review this manuscript.

The manuscript presents a rigorous study examining which clinical studies had been registered in 32 different registries. The manuscript presents the planned size, planned blinding, and whether mortality was a planned outcome for the identified planned studies. The authors suggest that the results of the study can be used to plan future studies and set research priorities. Unfortunately, considering the rapidly evolving situation I am afraid the findings might already be outdated. Additionally, several regularly updated alternatives exist, e.g. Cochranes “Covid-19 Study Register” (https://covid-19.cochrane.org/?pn=1), EBM Datalabs “Covid-19 TrialsTracker” (http://covid19.trialstracker.net/), and COVID-Evidence (https://covid-evidence.org/database).

Thus, I am afraid the reviewed manuscript will not be able to contribute substantially to priority setting or planning of future research activities: I do, however, find the methods to be technically sound and that the data supports the conclusions. My comments to the manuscript are presented below:

Author response: Thank you for thoroughly revising our manuscript. We agree that this study presents an overview of a dynamic research area, and in order to present the most relevant results, we have updated the manuscript with a new search on June 23, 2020.

Major issues:

Introduction:

• Several resources that track planned and completed trials are available online (including, but not limited to, the ones mentioned above). These should be mentioned in the introduction and it should be clarified how the presented study contributes.

Author response: The meta-overview of trial databases provided by The Global Health Network that sums up all these resources has been added as reference. We understand the concern regarding contribution, and much has changed since the first submission May 8. However, many resources transcript the ICTRP WHO-file without including registry entries from all 33 trial registries. In the current review, we assess outcome variables that are not easy to obtain from either of the databases available for download. Regarding contribution, we have revised: 

“Even though multiple versions of trial registration databases are readily available for download online [10-12], trialists may find it difficult to get an overview of the constantly evolving multitude of planned or ongoing studies. Therefore, we aim to provide a global snapshot overview of interventions and main methodological aspects of planned and initiated randomised clinical trials on COVID-19 prevention and treatment. To do so, we assess available trial registry entries from 33 clinical trial registries to June 23, 2020.” (page 3, line 20-26).

• Throughout the manuscript, the authors refer to protocols. A clinical study protocol is a quite elaborate document, that is e.g. submitted to ethical committees or institutional review boards to get ethical approval. Protocols are sometimes, but not always, published, but would not necessarily be available from trial registries, so it is my impression that the authors are in fact referring to trial registry entries. If the authors have in fact obtained protocols for all included trials, they should explain how these were obtained. Otherwise, I suggest changing the wording from protocol to trial registry entry throughout the text.

Author response: We apologize for the mix up in terminology. We have revised the text accordingly using the term “trial registry entries” throughout the text.

Methods:

• It is not explained why blinding is considered of particular importance. I would probably expect blinding to be less important for relatively objective outcomes such as overall mortality (although blinding could be very important for cause-specific mortality). Perhaps the authors could elaborate on the choice of outcome.

Author response: We agree that blinding is less important for mortality outcome, this is also supported by the literature (Anthon et al 2018 https://www.sciencedirect.com/science/article/abs/pii/S0895435617313501). However, we believe that the assessed parameters (blinding, use of mortality outcomes, trial size and multicentric designs) are individual contributors to better trial designs according to contemporary standards. Many other outcomes than mortality, including subjective outcomes, are being tested in the included trials, where lack of blinding can contribute to exaggerated interventions effects. In COVID-19 we have seen a much more liberal approach to implementation of new treatment modalities, and therefore believe that information on blinding is relevant. We have added to the Discussion: 

“Though there is no firm evidence that lack of blinding affects estimates of mortality [17], we consider blinding as an important factor in COVID-19 trial designs. Especially as 79% of trial registry entries had other primary outcomes than mortality and many were preventive with subjective and patient-reported outcomes such as self-assessed symptom severity where bias, including lack of blinding, that can exaggerate intervention effects [18].” (page 13, line 18-22).

• Additionally, in the results, the authors use the categories “Open label, single blind, double blind, triple blind, and quadruple blind”. It is not clear what these different categories describe, and research has shown that e.g. “double blind” is an ambiguous term [1,2]. The authors should explain how they operationalise blinding in the methods section.

Author response: We agree that the terminology regarding blinding is ambiguous. Information regarding which parties were blinded were not available from all trial registries. Hence, we merely counted the number of parties blinded to provide an overview of trial methodology. We have revised to “No. of parties blinded” in the tables. In the method section we added: 

“We reported the number of blinded parties registered in the trial registry entries, but not who was blinded.” (page 4, line 29-30).

• Regarding mortality as an outcome: Are the authors only looking at overall mortality or also cause-specific? This should be elaborated on.

Author response: We added to Methods: 

“Mortality was extracted binary regardless of cause-specificity or timely differences.” (page 4, line 27-28).

Discussion:

• Perspective: as mentioned above, I am afraid that the results of the study may already be outdated and more up-to-date resources exist.

Author response: We agree with the author that this is a dynamic research field. We have therefore added to our discussion: 

“This is a dynamic and rapidly developing research field and with the present study, we only provide a snapshot overview of trial registry entries of June 23, 2020. We however, believe that the overview of investigated interventions and quantification of several methodological parameters, not provided elsewhere, have merits. More adaptable inventories to find trial registry entries of specific interventions are available elsewhere [11, 12, 27].” (page 15, line 4-8). 

Minor issues:

Throughout the text the terms blinding, and masking are used interchangeably. If the authors believe these words are not describing the same this should be elaborated on, otherwise the text should be revised so only one term is used.

Author response: Thank you for your comments, we have revised the text accordingly using the term blinding throughout the manuscript and tables.

Abstract

• In conclusions the authors write: “a second wave of higher level evidence” – it is unclear what the first wave was and thus what “higher” is relative to. This is explained in the introduction but is not mentioned in the abstract. Thus, I suggest rephrasing.

Author response: Thank you for your comment, the abstract has been revised accordingly: 

“An extraordinary number of randomized clinical trials investigating COVID-19 management have been initiated with a multitude of medical preventive, adjunctive and treatment modalities.” (page 15, line 23-24).

Background:

• First paragraph: the authors write: “symptoms ranging from mild symptoms of upper airway infection to …” – I believe many reports state that a high proportion of people infected with SARS-CoV-2 are completely asymptomatic. Perhaps the authors could consider mentioning this.

Author response: Thank you for your comment. We agree and have revised as suggested and changed to a more updated reference: 

“… ranging from asymptomatic cases and mild symptoms of upper airway infection to fulminant acute respiratory distress syndrome (ARDS), multi-organ failure and death [1].” (page 3, line 3-5).

• Second paragraph: the authors write: “… to ensure a second wave of high-quality evidence with proper assessment of safety and efficacy measures” – I would suggest avoiding the term “safety” as this term tends to underplay harms and convey the idea that drugs have no, or non-important, side effects. I would suggest using “proper assessment of benefits and harms” – however, I am aware that this is not universally agreed upon, so just a suggestion.

Author response: Thank you for your comment. We agree and have revised as suggested: 

“… to ensure a second wave of high-quality evidence with proper assessment of benefits and harms” (page 3, line 16-17).

Methods:

• Data handling: The authors refer to “trial phase” – it is not immediately clear to me what they are referring to here – I assume it is the phases of clinical research, but some trials have multiple phases (e.g. double-blind and open-label), so perhaps the authors could elaborate a bit.

Author response: Thank you for your comment. We agree and have revised the term to “clinical trial phase” throughout the manuscript and tables. Further we elaborated in the method section: 

“The clinical research phase was registered (phase 1-4 trials) and whenever an entry had multiple trial phases, we registered the highest.” (page 4, line 27-29).

Results:

• First paragraph: The authors exclude a high proportion of identified studies; I would suggest mentioning the main reasons for exclusion, although these are also available from the PRISMA flowchart

Author response: Thank you for your comment. We agree and have added: 

“The vast majority of exclusions were due to single-group, observational and non-randomised designs, duplicates and testing of other interventions than COVID-19 treatment or prevention.” (page 6, line 3-5).

• Table 1 and Table 2: I suggest indenting the different types of interventions under “All trials” to make the table more legible.

Author response: Thank you for your comment, we agree and have revised the text accordingly in Table 1-3.

• Table 4a: The recruiting / completed variable is somewhat confusing. Completed could easily be interpreted as “trial completed” rather than “recruiting completed”.

Author response: Thank you for your comment, we agree and have changed the term to “Recruitment ongoing” and “Recruitment completed”.

Discussion:

• First paragraph: The first sentence is perhaps a bit too strongly worded, 770 RCTs were registered on the trial registries searched, there might be trials registered elsewhere.

Author response: We revised: 

“From January 23 to June 23, 2020 a total of 1303 randomized clinical trials were registered on the 33 trial registries searched, investigating 381 therapeutics or adjunct therapies in treatment of COVID-19.” (page 13, line 2-3).

• Second paragraph, second line: delete “from” in “ongoing trials initiated from before the pandemic”.

Author response: Revised accordingly.

• Second paragraph, fourth line: Rephrase the following sentence “China published 30% of the protocols”. The trials were not registered by China but from China. Also again, these are not published protocols but registry entries.

Author response: Thank you for your comment, after the update, this does no longer apply for registry entries from China and therefore has been deleted. 

However, we now find that 85% of completed recruitments were conducted in Iran, maybe because they still plan trials with smaller sample sizes. This was added to the discussion: 

“Entries in the Iranian Registry of Clinical Trials (IRCT) accounted for 15% of all registered RCTs but only 4% of all planned participants. Maybe because of the generally smaller trial sizes in entries from the IRCT, this registry contributed with 85% of all trials that had completed recruitment per June 23.” (page 13, line 14-17).

• Perhaps mention that such initiatives exist, e.g. the Living Meta-analysis from Cochrane France and others: https://covid-nma.com/

Author response: we have revised: 

“Collaboration to compile evidence in meta-analyses and network meta-analyses in order to achieve higher levels of evidence has been initiated, such as The Living mapping and living systematic review of Covid-19 studies [19-20].” (page 13, line 25-27). 

• Strengths and limitations: It is not mentioned in “methods” that data was extracted in duplicate.

Author response: Revised accordingly: 

“Trial registry entries were screened for inclusion and data were extracted by two authors independently (KZR or APHK and JL or OM)” (page 4, line 13-15.

• Strengths and limitations: The authors use the term “quality parameters”. Firstly, it is unclear to me what this refers to, I suppose blinding but that is only one parameter. Secondly, I would suggest avoiding the term “quality” which is somewhat normative and just call it blinding status instead.

Author response: Revised accordingly: 

“… and assessment of blinding status are strengths.” (page 14, line 28).

• Strengths and limitations: I am unsure why including “unconventional medicine” would be controversial, it is an important part of the narrative of the research being conducted.

Author response: We appreciate your comment and have deleted this section.

• Supplementary information: Appendix 2 and appendix 3 are called “Excluded articles” and “included articles”, however the unit of analysis is not articles but registry entries.

Author response: Revised accordingly

 

References

1 Hróbjartsson A, Pildal J, Chan A-W, et al. Reporting on blinding in trial protocols and corresponding publications was often inadequate but rarely contradictory. J Clin Epidemiol 2009;62:967–73. doi:10.1016/j.jclinepi.2009.04.003

2 Devereaux PJ, Manns BJ, Ghali WA, et al. Physician Interpretations and Textbook Definitions of Blinding Terminology in Randomized Controlled Trials. JAMA 2001;285:2000–3. doi:10.1001/jama.285.15.2000

 

Reviewer #2: 

This manuscript describes the results of a systematic search for registered drug trials in the WHO trial registry that assess prevention or treatment for COVID-19. This is important to provide an overview of ongoing research, as well as its characteristics and quality. The authors assess key components of the trials, such as number of participants, outcome measures, blinding, status for recruiting, etc.

Overall, the methods and results are clearly presented and the authors use blinded data-extraction, a systematic search strategy, present a PRISMA flow-chart with reasons for exclusions, etc.

My main concern is that the authors have likely not, as stated in the manuscript, assessed the actual protocols for the trials. Trial registries generally do not included detailed protocols but a set of key pieces of information, which is a pity. Lots of important information is missing that would allow a more thorough assessment of the quality of the ongoing trials than just binding, e.g. by applying the Cochrane risk of bias tool and assess whether the randomisation process was adequate and likely to produce comparable groups. Given that I am correct about this, it might be necessary to change the terminology of the manuscript and explain that it is an evaluation of clinical trial registry forms rather than protocols and perhaps even use this to push for requirements to publish the full protocol along with the trial registry forms.

Author response: Thank you for revising our manuscript. It is correctly observed that the review is based on trial registry entries rather than full information protocols, that are not always linked in the registries. We apologize for the mix up in terminology between protocols and trial registry entries. We have changed “protocols” to “trial registry entries” throughout the manuscript. 

A full protocol overview would definitely have merits but were seldom available from the trial registries. We added to the limitation section: 

“This is a dynamic and rapidly developing research field and with the present study, we only provide a snapshot overview of trial registry entries of June 23, 2020. We however, believe that the overview of investigated interventions and quantification of several methodological parameters have merits. More adaptable inventories for finding trial registry entries of specific interventions are available elsewhere [11, 12, 27]. We reviewed trial registry entries and not actual trial protocols, mainly because these were not always available. Evaluation of trial protocols could have provided more detailed information on trial methodology and outcomes and we endorse recent suggestions of public access to all trial protocols concomitant with trial registration [28]” (page 15 line 4-11).

---

## [Decision Letter · Decision Letter 1]

6 Aug 2020

A systematic review of trial registry entries for randomized clinical trials investigating COVID-19 medical prevention and treatment

PONE-D-20-13626R1

Dear Dr. Karlsen,

We’re pleased to inform you that your manuscript has been judged scientifically suitable for publication and will be formally accepted for publication once it meets all outstanding technical requirements.

Kind regards,

Lisa Susan Wieland

Academic Editor

PLOS ONE

Additional Editor Comments (optional):

The comments of both reviewers have been addressed and the manuscript is much improved.

Reviewers' comments:

Reviewer's Responses to Questions

**Comments to the Author**

1. If the authors have adequately addressed your comments raised in a previous round of review and you feel that this manuscript is now acceptable for publication, you may indicate that here to bypass the “Comments to the Author” section, enter your conflict of interest statement in the “Confidential to Editor” section, and submit your "Accept" recommendation.

Reviewer #1: All comments have been addressed

2. Is the manuscript technically sound, and do the data support the conclusions?

Reviewer #1: Yes

3. Has the statistical analysis been performed appropriately and rigorously? 

Reviewer #1: Yes

4. Have the authors made all data underlying the findings in their manuscript fully available?

Reviewer #1: Yes

5. Is the manuscript presented in an intelligible fashion and written in standard English?

Reviewer #1: Yes

6. Review Comments to the Author

Reviewer #1: The authors have answered all comments in detail - and have updated their search. Thus I am happy to recommend this paper for publication in PLoS One

7. PLOS authors have the option to publish the peer review history of their article (what does this mean?). If published, this will include your full peer review and any attached files.

Reviewer #1: **Yes: **Asger Sand Paludan-Müller

---

## [Editor Report · Acceptance letter]

11 Aug 2020

PONE-D-20-13626R1 

A systematic review of trial registry entries for randomized clinical trials investigating COVID-19 medical prevention and treatment 

Dear Dr. Karlsen:

I'm pleased to inform you that your manuscript has been deemed suitable for publication in PLOS ONE. Congratulations! Your manuscript is now with our production department. 

Kind regards, 

on behalf of

Dr. Lisa Susan Wieland 

Academic Editor

PLOS ONE